# Does Bovine Raw Milk Represent a Potential Risk for Vancomycin-Resistant Enterococci (VRE) Transmission to Humans?

**DOI:** 10.3390/antibiotics14080814

**Published:** 2025-08-08

**Authors:** Elisa Massella, Simone Russo, Anita Filippi, Chiara Anna Garbarino, Matteo Ricchi, Patrizia Bassi, Elena Toschi, Camilla Torreggiani, Giovanni Pupillo, Gianluca Rugna, Valentina Carta, Cristina Bertasio, Andrea Di Cesare, Tomasa Sbaffi, Giulia Borgomaneiro, Andrea Luppi

**Affiliations:** 1Istituto Zooprofilattico Sperimentale della Lombardia e dell’Emilia Romagna, Strada Faggiola 1, Gariga di Podenzano, 29027 Piacenza, Italyandrea.luppi@izsler.it (A.L.); 2Istituto Zooprofilattico Sperimentale della Lombardia e dell’Emilia Romagna, Via Pietro Fiorini 5, 40127 Bologna, Italy; patrizia.bassi@izsler.it (P.B.);; 3Istituto Zooprofilattico Sperimentale della Lombardia e dell’Emilia Romagna, Strada dei Mercati 13a, 43126 Parma, Italy; camilla.torreggiani@izsler.it; 4Istituto Zooprofilattico Sperimentale della Lombardia e dell’Emilia Romagna, Via Pitagora 2, 42124 Reggio Emilia, Italy; 5Istituto Zooprofilattico Sperimentale della Lombardia e dell’Emilia Romagna, Via Emilio Diena 16, 41122 Modena, Italy; gianluca.rugna@izsler.it; 6Istituto Zooprofilattico Sperimentale della Lombardia e dell’Emilia Romagna, Via Antonio Bianchi 7/9, 25124 Brescia, Italy; valentina.carta@izsler.it (V.C.);; 7National Research Council of Italy-Water Research Institute (CNR-IRSA) Molecular Ecology Group (MEG), 28922 Verbania, Italy

**Keywords:** raw milk, *Enterococcus* spp., vancomycin resistance, *vanC* genes, whole genome sequencing

## Abstract

**Background/Objectives**: Vancomycin-resistant enterococci (VRE) are significant nosocomial pathogens worldwide, potentially transmitted by food-producing animals and related products. This study investigates the epidemiological role of bovine raw milk in the transmission of VRE to humans. **Methods**: Bulk milk samples were screened for *van* gene presence using a multiplex PCR. Mastitogenic enterococci isolated from individual milk samples were tested for antimicrobial susceptibility using the broth microdilution method. Strains not susceptible to vancomycin were whole genome sequenced. **Results**: Overall, *vanC* genes were detected in 299/1026 (29.14%) bulk milk samples. Specifically, *vanC1* was found in 204 samples (19.88%) and *vanC2/3* in 57 samples (5.56%), with both detected simultaneously in 38 samples (3.70%). Clinically significant *vanA* and *vanB* genes were not identified. A total of 163 mastitogenic *Enterococcus* strains were isolated from individual milk samples. Eight different *Enterococcus* species were detected, with *E. faecium* (104/163, 63.80%) and *E. faecalis* (34/163, 20.86%) being the most common. Multidrug resistance was observed in 106/163 (65.03%) isolates. The most common resistance frequencies were to ciprofloxacin and erythromycin (102/163, 62.58% both), followed by quinupristin/dalfopristin (93/163, 57.06%), linezolid (65/163, 39.88%), tetracycline (58/163, 35.58%), daptomycin (46/163, 28.22%), chloramphenicol (33/163, 20.25%), ampicillin, tigecycline, and high-dosage gentamycin (8/163, 4.91% all). Resistance to teicoplanin was not observed. Two vancomycin non-susceptible strains were identified: one *vanC2/3 E. casseliflavus* and one *vanC1 E. gallinarum*. Whole genome sequencing confirmed the presence of the complete *vanC* gene cluster and several virulence genes in both strains. **Conclusions**: Our findings suggest that while raw milk is unlikely to be a source of vancomycin resistance genes of highest clinical importance (*vanA* or *vanB*), it may contribute to the spread of *vanC* enterococci, which are increasingly associated with human infections.

## 1. Introduction

Enterococci are natural components of the human and animal microbiota, primarily located in the gastrointestinal tract, as well as in the oral and genital mucosae [1,2]. They are also commonly detected in the environment, including water, soil, and plants [3,4,5]. The genus *Enterococcus* (*E*.) comprises over 50 species, with *E. faecium* and *E. faecalis* being the most representative [6]. Although typically considered commensals, enterococci can act as opportunistic pathogens under favourable conditions. Globally, enterococci is one of the leading causes of nosocomial infections, being associated with considerable mortality, morbidity, and growing costs for the healthcare system [7]. These bacteria are responsible for a broad spectrum of infections, including genitourinary infections, bloodstream infections, endocarditis, peritonitis, meningitis, and neonatal sepsis [6], with high mortality rates (up to 20%) [8].

The clinical relevance of enterococci is underscored by their intrinsic resistance to a wide range of antimicrobial agents, including aminoglycosides, cephalosporins, macrolides and sulphonamides [9]. Concerningly, over the past decade, enterococci have increasingly acquired multidrug-resistant (MDR) profiles, including Critically Important Antimicrobials (CIAs).

In particular, vancomycin is one of the last-line therapeutic options for severe infections caused by enterococci and other MDR Gram-positive bacteria [7]. In recent years, the prevalence of vancomycin-resistant enterococci (VRE) has increased concerningly in the European Union (EU), including in the community setting.

Vancomycin resistance has become particularly widespread in *E. faecium*, the primary pathogenic species of the genus *Enterococcus* responsible for clinical infections in humans [10,11]. In 2021, vancomycin-resistant *E. faecium* was reported in 39% of the countries in Europe, with prevalence rates ranging from 25% to 66% [10,12]. In Italy, the incidence of clinical vancomycin-resistant *E. faecium* has shown a significant increase, rising from 11.1% in 2015 to 32.5% in 2023 [10]. Therefore, *E. faecium* has been included in the “WHO global priority pathogens list of antibiotic-resistant bacteria to guide research, discovery, and development of new antibiotics” [13]. Additionally, the spread of vancomycin resistance among enterococci also raises concerns about potential horizontal transfer of antimicrobial resistance gene (ARG) to other pathogens, such as *Staphylococcus aureus* [14,15]. Notably, clinical isolates of the *Enterococcus* genus also showed high resistance rates (>50%) to other critical antimicrobials, such as aminopenicillins and high-dose gentamicin, despite these trends having remained relatively stable in recent years [10,11].

Food-production animals and related products are strongly suspected of playing a role in the transmission of VRE to humans. Historically, the emergence of vancomycin resistance has been linked to the widespread use of similar molecules (such as avoparcin) as growth promoters in livestock until the late 1990s. Given the potential antimicrobial resistance (AMR) transmission through the food chain, the EU banned the use of avoparcin in animal husbandry in 1997 [16,17].

Currently, the potential transmission of VRE from livestock and related products to humans is monitored through the European surveillance programme (Commission Implementing Decision 2020/1729/EU), targeting vancomycin-resistant *E. faecalis* and *E. faecium* in selected meat-producing animals (broiler chickens, laying hens, fattening turkeys, and cattle under one year of age). The presence of enterococci in meat may be associated with faecal carcass contamination during critical slaughtering procedures (i.e., skinning or evisceration) or with faecal contamination of the environments where meat is processed [18].

However, other livestock sectors, such as dairy production, may also contribute to the transmission of critical resistances to humans.

Milk is considered a primary necessity food, with European production reaching 145 million tonnes in 2023, 8.9% of which is attributed to the Italian dairy industry [19]. The microbiological safety of this product is essential to protect consumers, as milk has already been associated with bacterial infections transmitted through raw dairy product consumption, including *E. coli* O157 [20,21,22].

Additionally, in the last decade, dairy cattle and related products have been identified as potential food sources of VRE [23,24,25,26,27,28]. Enterococci are known causative agents of both clinical and subclinical mastitis in cattle and small ruminants, typically originating from contaminated environments [29]. Once the mammary gland is colonised, enterococci can proliferate in the cistern and mammary ducts and be excreted in high concentrations in milk [27]. Additionally, faecal contamination during milking, especially under poor hygiene conditions, may further compromise milk quality [30].

Enterococcal infections in dairy cows may pose an occupational hazard for breeders and veterinarians and also represent a potential threat to individuals in close contact with livestock, especially if immunocompromised. Moreover, the growing consumer demand for unpasteurised milk and artisanal cheeses, often driven by preferences for “natural” or minimally processed foods, represents a risk of foodborne transmission [31].

The lack of data on foodborne or occupational infections in humans associated with milk-origin VRE—likely due to underreporting and underestimation—highlights the importance of further investigating this issue.

This study aims to investigate the epidemiological role of raw bovine milk in the potential transmission of VRE and/or enterococci associated with other critical resistance profiles to humans. The phenotypic and genotypic characterisation of vancomycin resistance and virulence determinants in milk-derived VRE strains will provide valuable insights into the mechanisms of pathogenicity, therapeutic challenges, and the epidemiology of VRE infections.

## 2. Results

### 2.1. Bulk Milk

#### *van* Gene Distribution

*van* genes were detected in 299/1026 (29.14%) bulk milk samples. Specifically, *vanC1* was identified in 204 samples (19.88%), *vanC2/3* in 57 samples (5.56%), and both *vanC1* and *vanC2/3* in 38 samples (3.70%). *vanA* and *vanB* genes were not detected.

### 2.2. Individual Milk

#### 2.2.1. *Enterococcus* Species Distribution

A total of 163 *Enterococcus* spp. strains were isolated, belonging to eight different species. The most common were *E. faecium* (104/163, 63.80%) and *E. faecalis* (34/163, 20.86%), followed by *E. hirae* (9/163, 5.52%), *E. cecorum* (6/163, 3.68%), *E. saccharolitycus* (4/163, 2.45%), *E. canintestini* (3/163, 1.84%), *E. casseliflavus* (2/163, 1.23%), and *E. gallinarum* (1/163, 0.61%) (Table 1).

#### 2.2.2. AMR Profiles of *Enterococcus* spp. Strains

A total of 158 out of 163 (96.93%) strains were non-susceptible to at least one antimicrobial, meanwhile 106/163 (65.03%) isolates were MDR from three up to nine different antimicrobial classes. Most strains showing non-susceptibility to at least five different antimicrobial classes belonged to *E. faecium* (26/33, 78.79%) (Table 1).

The highest percentages of non-susceptibility strains resulted in ciprofloxacin and erythromycin (102/163, 62.58% both), quinupristin/dalfopristin (93/163, 57.06%), linezolid (65/163, 39.88%), tetracycline (58/163, 35.58%), daptomycin (46/163, 28.22%), chloramphenicol (33/163, 20.25%), ampicillin, tigecycline, and high-dosage gentamycin (8/163, 4.91% both). All strains were susceptible to teicoplanin (Figure 1).

Interestingly, two strains were non-susceptible to vancomycin: one *E. casseliflavus* (MIC = 8 μg/mL) and one *E. gallinarum* (MIC = 16 μg/mL), carrying *vanC2/3* and *vanC1*, respectively. Both strains showed MDR profiles, being also resistant to quinupristin/dalfopristin and linezolid (for both *E. casseliflavus* and *E. gallinarum*), and ciprofloxacin (for *E. casseliflavus*) and daptomycin (for *E. gallinarum*).

#### 2.2.3. WGS of VRE Strains

Both *vanC2/3 E. casseliflavus* and *vanC1 E. gallinarum* harboured the entire *vanC* gene cluster, characterised by five genes, namely *vanC*, *vanXY*, *vanT*, *vanR*, and *vanY*. Different virulence genes (VAGs) involved in pathogenesis were also detected and reported in Table 2.

### 2.3. Statistical Analysis Results

Both *E. faecium* and *E. faecalis* demonstrated high multidrug resistance (MDR) rates, with 73.08% (76/104) and 67.65% (23/34), respectively. Nevertheless, the MDR profile did not show a significant association with either species. Pearson’s Chi-square test revealed significant differences in antimicrobial resistance (AMR) patterns between the two species. Specifically, resistance to quinupristin/dalfopristin and tetracycline was significantly more frequent in *E. faecalis* compared to *E. faecium* (*p* < 0.001), whereas *E. faecium* showed significantly higher resistance to daptomycin, ciprofloxacin, and linezolid (*p* < 0.001). No statistically significant differences in resistance to other antimicrobials were observed between the two species, based on Pearson’s Chi-square or Fisher’s exact tests.

## 3. Discussion

Enterococci, although commensals of the human gut, can cause serious opportunistic infections such as urinary tract infections, bacteremia, and endocarditis [6]. Their intrinsic resistance to different antimicrobials, along with a high capacity to acquire new ARGs, makes them clinically significant [32]. Particularly concerning is the increasing detection of VRE, considered major nosocomial pathogens worldwide [11]. Given the rising incidence of VRE infections and the limited availability of effective treatments, significant research has been focused on identifying VRE reservoirs and transmission routes, including those beyond healthcare settings.

The food animal sector has long been suspected of playing a role in the dissemination of VRE to humans. While poultry, swine, and bovine meat have been extensively studied and monitored for VRE contamination [33,34,35,36,37,38,39,40,41,42,43,44,45,46], dairy products, particularly raw milk, and raw milk cheeses, have received comparatively less attention. However, these products may represent potential vectors for VRE transmission.

Indeed, raw milk can easily become contaminated with faecal enterococci during milking procedures due to inadequate hygienic practices, such as improper udder cleaning, use of contaminated equipment (e.g., teat cups, milking machines) or poor sanitation of milking parlour surfaces [47]. Moreover, the high tolerance of enterococci to adverse conditions [48,49], combined with their biofilm-forming ability that promotes strong surface adhesion [50], supports their prolonged environmental persistence and increases the risk of milk contamination during milking.

Vancomycin resistance screening revealed a high prevalence (29.14%) of *vanC* genes in bulk milk samples, while *vanA* and *vanB* genes were not detected. *vanC* genes are typically chromosomally encoded and primarily found in *E. gallinarum*, *E. casseliflavus*, and *E. flavescens*. They confer low level vancomycin resistance (MIC 4–32 mg/L) and susceptibility to teicoplanin [51]. Therefore, they are considered “less virulent” compared to the plasmid-encoded *vanA* and *vanB* genes, associated with high level of vancomycin resistance, MDR profile and most human outbreaks [52]. However, the past decade has witnessed an increasing number of *vanC* isolates associated with human disease [53,54,55,56,57,58,59,60,61,62,63,64,65]. This trend highlights the growing clinical relevance of *vanC* enterococci, even though *vanA* and *vanB* VRE remain the primary concern.

Despite the high prevalence of *vanC* genes detected in bulk milk samples, their occurrence in mastitic milk was limited to just two *Enterococcus* strains, identified as *vanC2/3 E. casseliflavus*, and *vanC1 E. gallinarum*. Once again, *vanA* and *vanB* genes were not identified in individual milk samples.

Notably, *vanC* strains were MDR to several critical antimicrobials, including vancomycin, quinupristin/dalfopristin, linezolid, daptomycin, and ciprofloxacin, challenging the long-standing assumption of VanC phenotypes as “low-virulent” pathogens. From a genomic perspective, both *vanC* strains harboured the whole *vanC* operon consisting of five genes (*vanC*, *vanXY*, *vanT*, *vanR*, and *vanS*) encoding for proteins involved in vancomycin resistance (VanC ligase, VanXY D, D-peptidase, and VanT serine racemase), as well as in the regulation of resistance gene expression (VanR/VanS two-component regulatory system). These findings are consistent with previous reports on vancomycin resistance gene clusters in these *Enterococcus* species [66]. Additionally, *vanC* enterococci were found to harbour a wide range of VAGs associated with key pathogenic mechanisms such as adhesion, immune evasion, biofilm formation, invasion, and surface protein anchoring. These genetic determinants, detailed in Table 2, likely contribute to the ability of *vanC2/3 E. casseliflavus* and *vanC1 E. gallinarum* isolated in this study to establish infection and cause mastitis.

Our data suggest that raw bovine milk and related products may serve as a source of emerging pathogenic *vanC* enterococci, while they are unlikely to represent a significant reservoir of the most clinically relevant *vanA* and *vanB* genes, as also reported by other authors [25,28,67,68,69].

Notably, mastitogenic enterococci from individual milk were associated with other critical resistances. *E. faecium* and *E. faecalis*, representing the most common *Enterococcus* species responsible for human infections [70], showed high MDR rate (*E. faecium*, 76/104, 73.08%; *E. faecalis*, 23/34, 67.65%), including to antimicrobials commonly used for enterococcosis treatments.

Ampicillin, high-dosage gentamycin/ampicillin association and vancomycin are considered recommended therapeutic options in case of enterococcal infections [71]. Resistance to these antimicrobials is routinely monitored by the European Antimicrobial Resistance Surveillance Network (EARS-Net). According to the last EARS Net-Annual Epidemiological Report, in 2023, more than nine-tenths (92.8%) of the invasive *E. faecium* isolates reported in the European countries were resistant to at least one of the antimicrobial groups under surveillance, meanwhile the percentage of high-level gentamicin resistance in *E. faecalis* was 24.3% [11]. Interestingly, in our study, resistance to high-dosage gentamycin and ampicillin was observed in nine and eight strains, respectively, mostly associated with *E. faecium* isolates. Notably, four *E. faecium* strains were resistant to both antimicrobials.

Treatment options for VRE infections are limited, with daptomycin and linezolid being the preferred first-line antibiotics [71]. Other antimicrobials such as quinupristin/dalfopristin and tigecycline might also be useful as salvage therapy [71], as well as fluoroquinolones and tetracycline in combination with other molecules in case of sepsis [72]. Concerningly, in this study, *Enterococcus* strains isolated from mastitic milk showed non-susceptibility to all aforementioned antimicrobials, with resistance rates ranging from 5% to 63%. The significant associations between *E. faecalis* and quinupristin/dalfopristin, tetracycline resistance and *E. faecium*, and ciprofloxacin and linezolid resistance were in accordance with previous results [73,74,75]. Linezolid resistance is typically reported as comparable between the two species [75,76,77,78] and our findings were consistent with this observation.

High resistance rates detected in mastitogenic enterococci are concerning as these bacteria could represent a source of AMR for humans. Mastitic milk is not intended for human consumption, as it may contain pathogens, potentially leading to foodborne illness and diffusion of AMR to humans, especially through the consumption of raw milk and related products [79,80]. In cases of severe clinical mastitis, abnormalities in the milk are easily detectable and the milk is typically discarded [81]. However, milk from cows with subclinical mastitis, where no visible changes are present, may inadvertently be mixed with bulk tank milk, entering the food chain and posing a potential hazard to consumers [82].

Furthermore, the excretion of mastitogenic AMR enterococci by infected cows can result in environmental contamination (especially of organic bedding materials, manure, wet or damp areas within barns or pastures and milking equipment) [83,84], which may subsequently lead to the contamination of milk intended for human consumption.

Therefore, mastitogenic milk may pose a risk to human health, either by directly contaminating milk intended for human consumption or by indirectly affecting the hygiene of the milking environment.

Preventing and controlling mastitis is therefore essential to ensure safe milk production and to reduce the spread of antimicrobial-resistant pathogens to humans [85].

Finally, our findings highlight the importance of molecular screening of *van* genes in bulk milk as an effective method for monitoring VRE in the dairy supply chain. Studies investigating the presence of *van* genes in bulk milk are limited and often focus solely on a narrow range of *Enterococcus* species, primarily *E. faecalis* and *E. faecium* [86,87,88,89]. Moreover, many of them rely exclusively on microbiological approaches, without considering molecular screening for *van* gene detection [86,87,88,89,90,91,92]. These limitations may hinder a comprehensive understanding of the dynamics of VRE transmission through raw milk, as evidenced by the absence or low detection rates of *vanC* enterococci reported in previous investigations. Conversely, molecular techniques offer high sensitivity and specificity, enable the rapid analysis of large numbers of samples, and provide valuable epidemiological insights into the role of raw milk in the transmission dynamics of VRE to humans.

## 4. Materials and Methods

### 4.1. Sample Collection

The study included bovine milk samples submitted to the IZSLER Laboratories in the Emilia Romagna region, Italy, between December 2022 and June 2024. Two types of samples were included in the study: (i) bulk milk samples, assessed for chemical parameters in accordance with the Italian payment programme based on milk quality, (ii) individual milk samples, analysed for routine mastitis prevention and control programmes. Samples were collected from dairy farms located in different Italian regions, predominantly in Emilia-Romagna, for a total of 1026 bulk milk samples and more than 80,000 individual milk samples. They were transported and stored at 4 °C until bacteriological and chemical evaluation. Information on the sampling procedures, farm treatments, and biosecurity measures in the farms included in the study was not provided by suppliers.

### 4.2. Van Gene Screening of Bulk Milk Samples

After homogenisation, 1 mL of each bulk milk sample was pre-enriched in 9 mL of Enterococcosel Broth (Oxoid), supplemented with 20 µg/mL vancomycin (Thermo Fisher Scientific, Waltham, MA, USA) and incubated at 37 ± 1 °C for 48 h. Genomic DNA was then extracted using the Biosprint 96 (Qiagen, Singapore) automated system with the MagMax CORE Nucleic Acid Purification Kit (Applied Biosystems, Thermo Fisher Scientific), following the manufacturer’s instructions. The presence of vancomycin resistance genes (*vanA*, *vanB*, *vanC1*, and *vanC2/3*) was investigated by multiplex PCR, following the protocol described by Dutka-Malen et al. [93].

### 4.3. Isolation and Identification of Enterococcus Isolates from Individual Milk Samples

Individual milk samples, tested for routine mastitis prevention and control programmes, were plated onto Aesculin Blood Agar (Oxoid) and incubated at 37 ± 1 °C for 24–48 h, following the National Mastitis Council protocols [29]. Aesculin-fermenting colonies were tested for Gram stain, catalase activity, and growth on Kanamycin Aesculin Azide Agar (Oxoid), incubated at 37 ± 1 °C for 24 h. Presumptive *Enterococcus* isolates were identified by matrix-assisted laser desorption/ionisation time-of-flight (MALDI-TOF) mass spectrometry.

#### 4.3.1. Antimicrobial Resistance Phenotyping and *Van* Gene Detection in *Enterococcus* Isolates

AMR profiles were evaluated by broth microdilution method on Sensititre^®^ EU Surveillance EUVENC plates (Thermo Fisher Scientific), according to the Clinical and Laboratory Standard Institute (CLSI) guideline [94]. Minimal inhibitory concentration (MIC) was assessed for 12 antibiotics, namely vancomycin, teicoplanin, quinupristin/dalfopristin, tetracycline, daptomycin, ciprofloxacin, erythromycin, tigecycline, linezolid, gentamicin, ampicillin, and chloramphenicol. The antimicrobial panel was proposed by Commission Implementing Decision 2020/1729/EU on the monitoring and reporting of AMR in zoonotic and commensal bacteria. MIC results were interpreted according to CLSI clinical breakpoints [94,95], or when not available, to the EUCAST clinical ones [96]. Strains were categorised as “susceptible” or “non-susceptible”, where the term “non-susceptible” encompasses both resistant and intermediate isolates, according to the definition of Sweeney et al. [97]. Isolates that were non-susceptible to at least three different antimicrobial classes were considered multidrug-resistant (MDR). Only acquired AMRs were considered to address a strain as MDR [98].

The AMR profiles of the whole collection are shown as a heatmap (Figure 1), depicting non-susceptibility (violet) or susceptibility (beige) to different antimicrobials. The dendogram on the left clusters the strains according to the similarities in their AMR profiles. This graphical representation was generated using R software version 3.5.1 (R Foundation for Statistical Computing) and RStudio version 1.1.463 (RStudio Inc., Boston, MA, USA). Finally, strains not susceptible to vancomycin (MIC ≥ 8 µg/mL) were tested for van gene presence by multiplex PCR [93].

#### 4.3.2. Whole Genome Sequencing (WGS) and Assembly

The whole genomes of VRE strains were sequenced to deeply analyse their genomic features. Bacterial isolates were suspended in 1 mL of physiological solution and genomic DNA was extracted using the Nucleospin Tissue kit (Machery-Naghel), according to the manufacturer’s instructions. DNA concentrations were quantified with the QuantiFluor^®^ ONE dsDNA System using the Quantus Fluorometer (Promega, Madison, WI, USA).

Genomic libraries were prepared using the Illumina DNA Prep (M) Tagmentation Kit (Illumina, San Diego, CA) and WGS was performed on the MiniSeq System (Illumina), generating 2 × 150 bp paired-end reads.

Raw reads were uploaded to the KBase platform [99]. Read quality was assessed using FastQC (version 0.12.1) (https://www.bioinformatics.babraham.ac.uk/projects/fastqc/, accessed on 12 February 2025), which evaluated average read length, duplication levels, and GC content distribution. Trimmomatic v0.36 [100] was employed to trim and filter raw reads using the following parameters: sliding window size = 4, sliding window minimum quality = 20, minimum read length = 50. Quality-filtered reads were assembled using SPAdes version 3.15.3 [101], specifying the kmer list (21, 33, 55, 77, 99) and a minimum contig length of 1000 bp. The quality of the assemblies was assessed using QUAST version 4.4 [102].

#### 4.3.3. Detection of Antimicrobial Resistance Genes, Virulence Factors, and Multi-Locus Sequence Typing (MLST)

Antimicrobial resistance genes were identified using the Resistance Gene Identifier tool (RGI, version 6.0.3, “main mode”) with the Comprehensive Antibiotic Resistance Database (CARD, version 4.0.0) [103]. The dataset included “strict” and “perfect” hits, while “loose” and “nudged” hits were excluded. Hits defined “nudged”, characterised by an identity > 95%, were reintroduced in the dataset only if the “percentage length of reference sequence” was >80%. Virulence genes were detected using the VFanalyzer pipeline (https://www.mgc.ac.cn/cgi-bin/VFs/v5/main.cgi, accessed on 12 February 2025) and the Virulence Factors Database (VFDB) (http://www.mgc.ac.cn/VFs/, accessed on 12 February 2025). Only hits that delivered an output with a gene name (rather than a numeric ID) were considered.

MLST analysis was performed using the PubMLST web interface (https://pubmlst.org/organisms, accessed on 12 February 2025) and the BIGSdb software [104] to characterise the isolates and determine their sequence types (STs). All the sequencing data of the *Enterococcus* isolates from this study are deposited in the National Centre for Biotechnology Information (NCBI) database as BioProject PRJNA1272009.

### 4.4. Statistical Analysis

Descriptive statistics (absolute frequencies and percentages) were performed considering (i) *van* gene distribution in bulk milk samples, (ii) enterococcal species, occurrence, and AMR characteristics (non-susceptibility to antimicrobials, MDR profiles) of *Enterococcus* spp. isolates from individual milk samples. Pearson’s Chi-square test or Fisher’s exact test was applied to compare the phenotypic AMR and MDR profiles of *E. faecalis* and *E. faecium* strains in order to identify statistically significant differences between the two species. Significance was set at a *p*-value < 0.001. Statistical analyses were performed with SPSS version 27 (IBM).

## 5. Conclusions

This study provides important insights into the potential role of raw milk in the dissemination of vancomycin-resistant enterococci, particularly those harbouring *vanC* genes.

Notably, the dairy supply chain does not seem to be involved in the transmission of vancomycin resistance genes of the highest clinical importance (i.e., *vanA* and *vanB* genes). However, a high prevalence of *vanC* genes was detected in bulk milk samples, as well as in two *Enterococcus* strains isolated from mastitic milk. These findings suggest that raw milk may represent a potential route of transmission for *vanC* enterococci, which have increasingly been implicated in human clinical infections and nosocomial outbreaks in recent years.

Furthermore, the high prevalence of MDR *E. faecium* and *E. faecalis* strains, showing resistance to critical antimicrobials such as linezolid, daptomycin, and high-dosage gentamicin, raises significant public health concerns. MDR strains from mastitic milk, particularly in subclinical infections, may accidentally contaminate milk intended for human consumption and consequently be transmitted to humans through dairy products.

Finally, the present study suggests how molecular-based methods are useful for VRE monitoring in the dairy supply chain. Such approaches show enhanced sensitivity and specificity compared to conventional microbiological methods and provide important epidemiological insight into the role of raw milk in the transmission of VRE to humans.

## Figures and Tables

**Figure 1 antibiotics-14-00814-f001:**
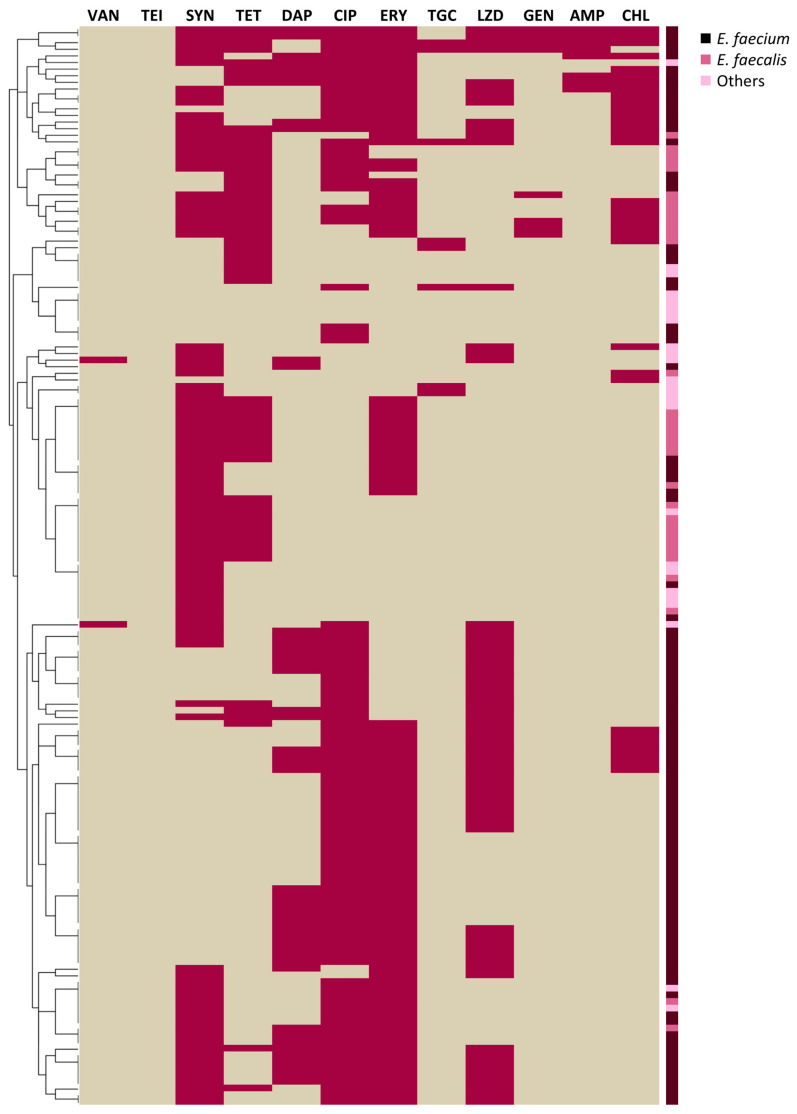
Heatmap depicting non-susceptibility (violet) or susceptibility (beige) to different antimicrobials of *Enterococcus* isolates, clustered according to the similarities of their antimicrobial resistance profiles. *Enterococcus* species are colour-coded. VAN: vancomycin, TEI: teicoplanin, SYN: quinupristin/dalfopristin, TET: tetracycline, DAP: daptomycin, CIP: ciprofloxacin, ERY: erythromycin, TGC: tigecycline, LZD: linezolid, GEN: high-dosage gentamycin, AMP: ampicillin, CHL: chloramphenicol.

**Table 1 antibiotics-14-00814-t001:** Number of isolates, number of isolates susceptible to all the antimicrobial panel tested (S), number of multidrug-resistant (MDR) strains, number of strains not susceptible to ≥5 different antimicrobial classes (≥5 R), according to *Enterococcus* species.

Species	N. of Isolates	S	MDR	≥5 R
*E. faecium*	104	-	76	26
*E. faecalis*	34	-	23	6
*E. hirae*	9	1	1	-
*E. cecorum*	6	4	-	-
*E. saccharolitycus*	4	-	1	-
*E. canintestini*	3	-	3	-
*E. casseliflavus*	2	-	2	-
*E. gallinarum*	1	-	-	-
Total	163	5	106	33

**Table 2 antibiotics-14-00814-t002:** Virulence genes (VAGs) and related functions carried by vancomycin non-susceptible *E. gallinarum* and *E. casseliflavus*.

*E. gallinarum*	
VAGs	Function
*ebpA*, *ebpB*, *ebpC*, *srtC*, *efaA*, *pavA*, *slrA*	Adherence
*cpsA/uppS*, *cpsB/cdsA*, *cpsI*, *cpsJ*	Antiphagocytosis
*bopD*	Biofilm formation
*cylR2*, *cesC*	Toxin
*htrA/degP*	Chemotaxis and motility, invasion
*cheY*	Regulation
*bscN*	Secretion system
*lgt*	Surface protein anchoring
*E. casseliflavus*	
VAGs	Function
*ebpA*, *ebpC*, *srtC*, *efaA*, *pavA*, *slrA*	Adherence
*cpsA/uppS*, *cpsB/cdsA*, *cpsI*, *cpsJ*, *uge*	Antiphagocytosis
*bopD*	Biofilm formation
*ctpV*	Copper uptake
*capO*, *capP*, *capD*, *cps4I*, *cpsY*, *epsE*,	Immune evasion
*htrA/degP*	Chemotaxis and motility, invasion
*cheY*	Regulation
*gspE*, *bscN*	Secretion system
*lgt*	Surface protein anchoring

## Data Availability

Dataset available on request from the authors.

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
