# Peer review of "Does Bovine Raw Milk Represent a Potential Risk for Vancomycin-Resistant Enterococci (VRE) Transmission to Humans?"

_antibiotics, 2025, doi:10.3390/antibiotics14080814_

Round 1

Reviewer 1 Report

Comments and Suggestions for Authors

This study offers a valuable contribution to the understanding of antimicrobial resistance in the dairy sector by systematically investigating the presence of vancomycin-resistant enterococci (VRE) in bovine milk. A notable strength lies in the large-scale sampling of over 1,000 bulk milk samples and 80,000 individual milk samples, which enhances the robustness and representativeness of the findings. The manuscript is well-written, clear, and logically structured. However, there are major flaws need to address.

The main claim is about the risk of VRE transmission from raw milk, but no vanA or vanB genes, the clinically critical types, were detected. The absence of vanA/vanB undermines the study’s implication that raw milk is a meaningful reservoir of clinically significant VRE.

The vanC genes found (vanC1, vanC2/3) are typically associated with low-level resistance and commensal enterococci, not high-risk clinical infections.The author mention that vanC-positive strains represent a serious threat to human health is overstated and speculative without direct epidemiological or clinical correlation.

The study does not analyze potential farm-level contributors (hygiene, antibiotic use, animal health) to VRE prevalence. Without farm-level data, no risk factor analysis is possible. This limits the study's practical application and epidemiological value.

Out of 80,000+ individual milk samples, only 2 strains (1 E. casseliflavus, 1 E. gallinarum) were found to be vancomycin non-susceptible. This weakens the validity of the risk assessment. The methodology should have been optimized or adapted to enrich clinically relevant VRE (e.g., E. faecium and E. faecalis carrying vanA/vanB).

The presence of van genes was reported qualitatively in bulk milk, but no quantification (e.g., Ct values, copy number, bacterial load). This makes it impossible to assess the risk level or public health relevance of gene presence.

While vanC genes were detected in bulk milk, no bacterial isolates were obtained from those samples. The study discusses bulk milk and mastitic milk separately, but does not analyze if farms positive for van genes in bulk milk also had mastitic cases with VRE.

Statistical analysis focuses mainly on species-level resistance differences, no regression or risk models. While the number of bulk milk samples (1026) is impressive, only two VRE strains (one each of E. casseliflavus and E. gallinarum) were isolated with vancomycin resistance. This raises questions about the representativeness and statistical power for assessing real transmission risk to humans. A control group or more stratified sampling (e.g., comparing farms with good vs. poor hygiene) would strengthen conclusions about risk factors.

In table 1, author mentioned about 114/163 isolates.  Author should clear about remaining isolates condition.

In Table 2, mention the section of E. gallinarum

Author did not clear about genes, how they detect? With primer list or not.

Author Response

Response to Reviewer 1 Comments

Comment 1: The main claim is about the risk of VRE transmission from raw milk, but no vanA or vanB genes, the clinically critical types, were detected. The absence of vanA/vanB undermines the study’s implication that raw milk is a meaningful reservoir of clinically significant VRE.

Response 1:

Thank you for pointing this out. The aim of our study was to evaluate the epidemiological role of raw milk in VRE transmission. According to our findings, raw milk does not appear to contribute to the dissemination of vanA and vanB enterococci, which are the most clinically significant types. However, a high percentage of bulk milk samples tested positive for vanC genes. Although vanC enterococci have historically been considered less problematic, infections caused by them have increased over the last decade, suggesting they may be emerging as potential pathogens.

Comments 2: The vanC genes found (vanC1, vanC2/3) are typically associated with low-level resistance and commensal enterococci, not high-risk clinical infections.The author mention that vanC-positive strains represent a serious threat to human health is overstated and speculative without direct epidemiological or clinical correlation.

Response 2: Thank you for pointing this out. Authors intent is not to overstated the epidemiological role of vanC enterococci, but to underlines their potential emerging role as pathogen, as witnessed by numerous case studies and reports associated with vanC enterococci infections. The authors tried to suggest that vanC enterococci could represent an emerging problem, but not the main one as described in lines 195-204, 223-223, 381-387.

Comments 3: The study does not analyze potential farm-level contributors (hygiene, antibiotic use, animal health) to VRE prevalence. Without farm-level data, no risk factor analysis is possible. This limits the study's practical application and epidemiological value.

Response 3: Thank you for pointing this out. Unfortunately, we did not have information regarding hygiene, antibiotic use or animal health on the farms included in the study. Therefore, we provided a general overview of the importance of biosecurity and proper hygiene practices, wich reflect on the animal health status and management of the breeding farms.

Comment 4: The presence of van genes was reported qualitatively in bulk milk, but no quantification (e.g., Ct values, copy number, bacterial load). This makes it impossible to assess the risk level or public health relevance of gene presence.

Response 4: Thank you for pointing this out. We agree with your comment. Unfortunately, we performed an end-poin PCR to detect the genes. Therefore, we weren’t able to quantify van gene in the samples (e.g., Ct values, copy number, bacterial load).

Comment 5: Out of 80,000+ individual milk samples, only 2 strains (1 E. casseliflavus, 1 E. gallinarum) were found to be vancomycin non-susceptible. This weakens the validity of the risk assessment. The methodology should have been optimized or adapted to enrich clinically relevant VRE (e.g., E. faecium and E. faecalis carrying vanA/vanB).

Response 5: Thank you for your comment. We tried our best to improve a methodology able to detect VRE, however, we didn’t find any protocol able to enrich clinically relevant VRE (e.g., E. faecium and E. faecalis carrying vanA/vanB).

Comment 6 : While vanC genes were detected in bulk milk, no bacterial isolates were obtained from those samples. The study discusses bulk milk and mastitic milk separately, but does not analyze if farms positive for van genes in bulk milk also had mastitic cases with VRE.

Response 6: Thank you for pointing this out. Our protocol does not include cultural examination of bulk tank milk for the isolation of environmental pathogens, but only contagious ones. Considering you second comment, unfortunately, we do not have specific information on these data, as bulk milk samples submitted for quality analysis are identified only by barcodes, whereas milk samples submitted for mastitis control are assigned numbers generated by our server. This discrepancy makes it very difficult to determine the presence of mastitis cases on farms with bulk milk samples testing positive for vanC genes.

Comment 7: Statistical analysis focuses mainly on species-level resistance differences, no regression or risk models. While the number of bulk milk samples (1026) is impressive, only two VRE strains (one each of E. casseliflavus and E. gallinarum) were isolated with vancomycin resistance. This raises questions about the representativeness and statistical power for assessing real transmission risk to humans. A control group or more stratified sampling (e.g., comparing farms with good vs. poor hygiene) would strengthen conclusions about risk factors.

Response 7: Thank you for your comment; we agree with your observations. Due to the low number of VRE isolates from mastitic milk and the inability to perform statistically robust analyses on VRE detection in individual milk samples, we decided to focus our statistical analysis on comparing the antimicrobial resistance profiles of mastitogenic E. faecalis and E. faecium, which were the most commonly identified species in the milk samples.

Comment 8: In table 1, author mentioned about 114/163 isolates.  Author should clear about remaining isolates condition.

Response 8: Thank you for your comment. Table 1 describes the number of resistance to different antimicrobial classes (lines 126-129). MDR isolates showed resistant to at least 3 different antimicrobial classes. Within this category, the authors wish to underline strains showing resistance to at least 5 different antimicrobial classes, representing a subcategory of MDR strains with a very high number of resistances. Considering E. faecium, we detected 104 strains. All the strains were resistant to at least one antimicrobial (S=0). Considering the 104 strains,  76 were MDR (resistant to at least 3 different antimicrobial classes). Of these 76 strains, 26 isolates belonged to the subcategory of MDR, showing resistance to at least 5 different antimicrobial classes. The missing strains (104-76 = 28) (not mentioned in the table) are the strains resistant to 1 or 2 different antimicrobial classes (so the strains between S category and MDR category.

Comment 9: In Table 2, mention the section of E. gallinarum

Response 9: Thank you for pointing this out. Done.

Comment 10: Author did not clear about genes, how they detect? With primer list or not

Response 10: Thank you for your comment. van genes detection and protocols are described in Material and methods, van gene screening of bulk milk samples, lines 300-302.

le

Reviewer 2 Report

Comments and Suggestions for Authors

This is a well-designed study addressing a significant public health concern: the role of bovine raw milk in transmitting vancomycin-resistant enterococci (VRE) to humans. The manuscript is well structured, methods are robust, and conclusions are largely supported by the data. There are some suggestion to improve the manuscript:

 Abstract:

“Vancomycin-resistant enterococci (VRE) are significant nosocomial pathogens worldwide, potentially transmitted by food-producing animals and related products.”

Nosocomial infection are hospital-acquired infection, so rather emphasizing nosocomial importance, we should mention zoonotic importance in the opening sentence of the manuscript.

Introduction:

The importance of milk borne pathogens may be emphasized in the introduction. Masititis is usually associated with different bacteria mainly Staphylococci, Streptococci, E. coli, Mycoplasma, Enterococci, Acinetobacter, Klebsiella etc.

So it may be better to introduce the importance of milkborne bacterial infections and then we should mainly focus on Enterococci.

Junjun Liu, Xinglin Zhang, Jianrui Niu, et al.,

Complete Genome of Multi-Drug Resistant Staphylococcus aureus in Bovine Mastitic Milk in Anhui, China
Pak Vet J, 2023, 43(3): 456-462

Sohad M Dorgham, Amany A Arafa, Eman S Ibrahim and Abeer M Abdalhamed

Carbapenem-resistant Acinetobacter baumannii in Raw Milk from Egyptian Dairy Farm Animals with Subclinical Mastitis

Int J Vet Sci, 2024, 13(6): 813-818.

Sadık SAVAÅžAN, ÇaÄŸatay NUHAY, Volkan Enes ERGÜDEN, Serap SAVAÅžAN.

Determination of Biofilm Formation, Antibacterial Resistance and Genotypes of Bacillus cereus Isolates from Raw Milk.

Kafkas Univ Vet Fak Derg, 29 (3): 265-271, 2023.

https://doi.org/10.9775/kvfd.2023.29162

Material and methods:

Selection of antibiotics: Selection criteria of 12 antibiotics may be added e.g. as per CLSI guidelines.

Identification / confirmation of different Enterococcus species may be included.

Results:

Table 1 & 2: Title should be above Table.

E. gallinarum name is missing: Line 161

267-271

2 line paragraphs may be avoided.

Excellent discussion on mastitis (subclinical vs. clinical) as a route for MDR enterococci transmission. Moreover,

We may also discuss “fecal contamination vs. intramammary infection”

Line 372: E. faecalis           italicize it

Abstract Conclusions: Our findings suggest that while raw milk is unlikely to be a source of vancomycin resistance genes of highest clinical importance (vanA or vanB), it may contribute to the spread of vanC enterococci, which are increasingly associated with human infections.

Line 199: they are considered “less virulent” compared

Can we say vanC is more prevenlant but less virulent?

Line 318: ampicillin e chloramphenicol.

Do you mean: ampicillin and chloramphenicol.

Italicize the bacterial species names in references too.

Author Response

Response to Reviewer 2 Comments

Comment 1: “Vancomycin-resistant enterococci (VRE) are significant nosocomial pathogens worldwide, potentially transmitted by food-producing animals and related products.” Nosocomial infection are hospital-acquired infection, so rather emphasizing nosocomial importance, we should mention zoonotic importance in the opening sentence of the manuscript.

Response 1: Thank you for pointing these out. We prefer to retain both concepts: that VRE are associated with nosocomial infections, and that they can potentially be transmitted by food animals and related products, thus having a zoonotic potential.

Comment 2: The importance of milk borne pathogens may be emphasized in the introduction. Masititis is usually associated with different bacteria mainly Staphylococci, Streptococci, E. coli, Mycoplasma, Enterococci, Acinetobacter, Klebsiella etc.

So it may be better to introduce the importance of milkborne bacterial infections and then we should mainly focus on Enterococci.

Response 2: Thank you for your comment. Done (lines 95-99)

Comment 3: Selection of antibiotics: Selection criteria of 12 antibiotics may be added e.g. as per CLSI guidelines.

Response 3: Thank you for pointing these out. The selection criteria of the antibiotics are described in the lines 317-320.  

Comment 4: Identification / confirmation of different Enterococcus species may be included.

Response 4: Thank you for pointing these out Identification/confirmation of different Enterococcus species is described in Materials and Methods, Isolation and identification of Enterococcus isolates from individual milk samples, lines 308-310.

Comment 5: Table 1 & 2: Title should be above Table.

Response 5: Thank you for pointing these out. Done

Comment 6: E. gallinarum name is missing: Line 161

Response 6: Thank you for your comment. Added.

Comment 7: 267-271 line paragraphs may be avoided.

Response 7: Thank you for pointing this out. We believe it is important to retain these lines in order to emphasize this fundamental concept

Comment 8: We may also discuss “fecal contamination vs. intramammary infection”

Response 8: Thank you for pointing this out. Faecal contamination is discussed in lines 188-194, meanwhile intramammary infection in lines 94-100.

Comment 9: Line 372: E. faecalis           italicize it

Response 9: Thank you for pointing this out. Done

 Comment 10: Line 199: they are considered “less virulent” compared. Can we say vanC is more prevenlant but less virulent?

Response 10: Yes, considering our study.

Comment 11: Line 318: ampicillin e chloramphenicol. Do you mean: ampicillin and chloramphenicol.

Response 11: Thank you for your comment. Yes, modified.

Comment 12: Italicize the bacterial species names in references too.

Response 12: Thank you for your comment. Done.

Reviewer 3 Report

Comments and Suggestions for Authors

The research is of interest as it focuses on a potential pathway for the spread of antibiotic resistance genes. The research approach and methods are appropriate.

Specific Comments:

Line 99. Please provide perspective on the access and sale of raw milk in Italy, EU, other regions of the world.  May provide readers with greater perspective of the issue associated with sale/consumption of raw milk or raw milk products.

Line 101: Please provide reference or surveillance/epidemiological data on infections/outbreaks with Enterococci linked to consumption of raw (non-pasteurized) milk. Also, documentation of cases of Enterococci illness among farm personnel milking cows, veterinarians, and/or farmers.

Is there any information on community acquired infections linked to exposure to raw milk.

Line 195: What was the incidence of mastitis caused by Enterococci on farms in which bulk tank samples were positive. None to low incidence may suggest the isolates were environmental origin and not associated with sub-clinical or clinical mastitis. I know that this is somewhat addressed on line 205-208. Wondering what percentage of cases of mastitis on those farms were linked to Enterococci infections.

Line 246: Were cattle treated with any of the antibiotics prophylactic or therapeutic?

Line 267: In this study is there any sense as to whether AMR enterococci were acquired from the environment or shed from infected cattle?

Line 284: Provide greater information on sample collection.  For example, for individual samples were the cow’s udder washed and teats dried prior to collection. Were teat ends swabbed with alcohol swab prior to expression of milk into the collection tube. Were several streams of milk expressed prior to collection of the sample. Were samples taken prior to milking or after removal of the milking machine. 

For bulk tank samples were samples collected from bulk tanks during milk when the milk temperature would higher than after several hours of cooling. For bulk tank samples on average how many milking or days passed before samples were collected. Or how many days had the milk been in the bulk tank prior to collection of bulk tank samples.

Author Response

Response to Reviewer 3 Comments

Comment 1: Line 99. Please provide perspective on the access and sale of raw milk in Italy, EU, other regions of the world.  May provide readers with greater perspective of the issue associated with sale/consumption of raw milk or raw milk products.

Response 1: Thank you for pointing this out. We agree with this comment. Therefore, we have added in lines 93-95.

Comment 2: Line 101: Please provide reference or surveillance/epidemiological data on infections/outbreaks with Enterococci linked to consumption of raw (non-pasteurized) milk. Also, documentation of cases of Enterococci illness among farm personnel milking cows, veterinarians, and/or farmers.

Response 2: Thank you for pointing this out. Unfortunately, to the best of our knowledge, there aren’t data on infections/outbreaks with Enterococci linked to consumption of raw (non-pasteurized) milk or documentation of cases of Enterococci illness among farm personnel milking cows, veterinarians, and/or farmers.

Comment 3: Is there any information on community acquired infections linked to exposure to raw milk.

Response 3: Thank you for your comment. Yes, there are,  but involved other bacteria not included in our study (lines 93-95).

Comment 4: Line 195: What was the incidence of mastitis caused by Enterococci on farms in which bulk tank samples were positive. None to low incidence may suggest the isolates were environmental origin and not associated with sub-clinical or clinical mastitis. I know that this is somewhat addressed on line 205-208. Wondering what percentage of cases of mastitis on those farms were linked to Enterococci infections.

Response 4: Thank you for pointing this out. Unfortunately, we do not have specific information on these data, as bulk milk samples submitted for quality analysis are identified only by barcodes. This makes it very difficult to determine the percentage of mastitis cases present in the farms submitting these samples.

Comment 5: Line 246: Were cattle treated with any of the antibiotics prophylactic or therapeutic?

Response 5: Thank you for your comment. Unfortunately, we don’t have information about these data.

Comment 6: Line 267: In this study is there any sense as to whether AMR enterococci were acquired from the environment or shed from infected cattle?

Response 6: Thank you for pointing this out. Unfortunately, it is not possible to establish if enterococci were acquired from the environment or shed from infected cow.

Comment 7: Line 284: Provide greater information on sample collection.  For example, for individual samples were the cow’s udder washed and teats dried prior to collection. Were teat ends swabbed with alcohol swab prior to expression of milk into the collection tube. Were several streams of milk expressed prior to collection of the sample. Were samples taken prior to milking or after removal of the milking machine. 

Response 7: Thank you for pointing this out.  Unfortunately, we don’t have information about sample collection, because we just received the samples to test, and we are not involved in the sampling. Our lab consistently recommends following proper hygiene practices, but we were unable to verify whether these recommendations were actually implemented.

Comment 8: For bulk tank samples were samples collected from bulk tanks during milk when the milk temperature would higher than after several hours of cooling. For bulk tank samples on average how many milking or days passed before samples were collected. Or how many days had the milk been in the bulk tank prior to collection of bulk tank samples.

Response 8: Bulk milk samples were usually analyzed on the day of collection or, at most, within 48 hours. Aniway, since we are not in charge for the sampling at any receivement of the samples we registrate the temperature of transport/storage, that should be between +2 and +8°C. After arrival samples were stored at 4°C until processing.

Reviewer 4 Report

Comments and Suggestions for Authors

I have reviewed the manuscript titled “ Does bovine raw milk represent a potential risk for Vancomycin Resistant Enterococci (VRE) transmission to humans?”.  Antibiotic resistance represents one of the most pressing and alarming public health concerns in the modern era. From the One Health perspective, antibiotic resistance represents one of the foremost priorities. Therefore, the manuscript not only provides information on a current and significant issue but is also expected to attract the interest of readers. However, certain improvements are necessary to enhance its clarity and compliance with academic standards.

-Please provide more detailed information about the sample in the abstract section.

-Please include brief information supported by recent literature regarding the use of vancomycin in the Introduction section of the manuscript.

-I recommend including a brief explanation of the “One Health” approach in the introduction section.

-It is recommended that all abbreviations be defined upon their first appearance in the manuscript, and the abbreviation alone be used thereafter.

-Please clearly include detailed information about the chemicals, kits, and equipment used in the manuscript.

-Kindly review and confirm the accuracy of the data presented on lines 33-34, and 143.

-Please review the references section to ensure its compliance with the journal’s guidelines.

Author Response

Response to Reviewer 4 Comments

Comment 1: Please provide more detailed information about the sample in the abstract section.

Response 1: Thank you for pointing this out. Unfortunately, we were not able to provide more information about samples in the abstract session because of the word limits. However, samples are described in Material and methods – Sample collection – lines 284-293.

Comment 2: Please include brief information supported by recent literature regarding the use of vancomycin in the Introduction section of the manuscript.

Response 2: Thank you for pointing this out. Information about the use of vancomycin has been provided in the Introduction, lines 80-82.

Comment 3: It is recommended that all abbreviations be defined upon their first appearance in the manuscript, and the abbreviation alone be used thereafter.

Response 3: Thank you for the comment. We have revised the manuscript accordingly.

Comment 4: Please clearly include detailed information about the chemicals, kits, and equipment used in the manuscript.

Response 4: Thank you for pointing this out. Detailed information about the chemicals, kits, and equipment used in the manuscript has been checked and provided in the Section Material and methods, lines 284 – 376.

Comment 5: Kindly review and confirm the accuracy of the data presented on lines 33-34, and 143.

Response 5: Thank you for pointing this out. We have checked accuracy and modified the data.

Comment 6: Please review the references section to ensure its compliance with the journal’s guidelines.

Reference 6: Thank you for pointing this out. Done.

Round 2

Reviewer 1 Report

Comments and Suggestions for Authors

The authors did not address well comments 3 to 7. Without addressing these issues, how we can proceed further.

Author Response

Response to Reviewer 1 Comments

Dear rewiever 1,

We are sorry we can’t properly answer your questions.

Comments 3: The study does not analyze potential farm-level contributors (hygiene, antibiotic use, animal health) to VRE prevalence. Without farm-level data, no risk factor analysis is possible. This limits the study's practical application and epidemiological value.

Response 3: Thank you for pointing this out. Unfortunately, we did not have information regarding hygiene, antibiotic use or animal health on the farms included in the study. Therefore, we provided a general overview of the importance of biosecurity and proper hygiene practices, which reflect on the animal health status and management of the breeding farms.

New response: considering comment 3, we agree with the Reviewer about the importance of these data, but

We agree with the Reviewer, this limit the epidemiological value of the study regarding the predisposing factors for VRE infection in dairy cow farms, but not the epidemiological role of milk involved in the potential transmission of VRE to humans. The aim of the study was to investigate the epidemiological role of raw bovine milk in the potential transmission of VRE and/or enterococci to humans and potential issues due to the resistance to antimicrobials of these strains. Unfortunately, we don’t have information on the sampling procedures, farm treatments and biosecurity measures about the farms involved in the study is not available because not provided by the suppliers. This was reported line 293–296 of the paper.

Comment 4: The presence of van genes was reported qualitatively in bulk milk, but no quantification (e.g., Ct values, copy number, bacterial load). This makes it impossible to assess the risk level or public health relevance of gene presence.

Response 4: Thank you for pointing this out. We agree with your comment. Unfortunately, we performed an end-point PCR to detect the genes. Therefore, we weren’t able to quantify van gene in the samples (e.g., Ct values, copy number, bacterial load).

New response: In response to comment 4, we were unable to quantify the van genes, but only to determine their presence or absence, according to the method used in this study. Nevertheless, the authors believe that vanC genes are carried by bacteria capable of replicating in milk, and that potentially even a single colony could pose a risk, particularly for immunocompromised individuals. Therefore, while we could not precisely assess the level of risk, we were able to confirm that a potential risk does exist.

Comment 5: Out of 80,000+ individual milk samples, only 2 strains (1 E. casseliflavus, 1 E. gallinarum) were found to be vancomycin non-susceptible. This weakens the validity of the risk assessment. The methodology should have been optimized or adapted to enrich clinically relevant VRE (e.g., E. faecium and E. faecalis carrying vanA/vanB).

Response 5: Thank you for your comment. We tried our best to improve a methodology able to detect VRE, however, we didn’t find any protocol able to enrich clinically relevant VRE (e.g., E. faecium and E. faecalis carrying vanA/vanB).

New response (comment 5): The validated protocol used in the lab for the isolation of bacterial mastitic agents, for the diagnosis of mastitis, does not include the use of selective media or enrichment procedures, to avoid results that do not reflect the real pathogenic role of the isolated microorganism. In fact the study has considered only Enterococci isolated in pure culture on non-selective media and therefore most likely associated to mastitis. 

Comment 6 : While vanC genes were detected in bulk milk, no bacterial isolates were obtained from those samples. The study discusses bulk milk and mastitic milk separately, but does not analyze if farms positive for van genes in bulk milk also had mastitic cases with VRE.

Response 6: Thank you for pointing this out. Unfortunately, we do not have specific information on these data, as bulk milk samples submitted for quality analysis are identified only by barcodes, whereas milk samples submitted for mastitis control are assigned numbers generated by our server. This discrepancy makes it very difficult to determine the presence of mastitis cases on farms with bulk milk samples testing positive for vanC genes.

New response: in response to comment 6, we would like to clarify that bulk milk samples and mastitic milk samples follow two separate processing paths. Bulk milk samples are identified only by a barcode, and the corresponding identification data are stored in dedicated programs that we do not routinely access. In contrast, mastitic milk samples are assigned identification numbers within our laboratory.

Both types of samples—those from farms submitting bulk milk and those related to mastitis—receive the results of the requested analyses (e.g., somatic cell count and bacterial count for bulk milk; identification of mastitis agents and corresponding antibiograms for mastitic milk) by an automatic transmission of them to the farmers. However, the results are stored on two separate servers.

Due to this separation, as well as the high volume of samples processed, it is extremely difficult to determine whether farms with vanC-positive bulk milk samples also have cases of mastitis associated with vanC enterococci. This also why there is a huge variability, in terms of frequency and number of samples, among the farms in sending samples for diagnostic purposes. Therefore, we are unable to provide this information.

Comment 7: Statistical analysis focuses mainly on species-level resistance differences, no regression or risk models. While the number of bulk milk samples (1026) is impressive, only two VRE strains (one each of E. casseliflavus and E. gallinarum) were isolated with vancomycin resistance. This raises questions about the representativeness and statistical power for assessing real transmission risk to humans. A control group or more stratified sampling (e.g., comparing farms with good vs. poor hygiene) would strengthen conclusions about risk factors.

Response 7: Thank you for your comment. The high prevalence of VRE in bulk milk suggest a potential transmission of vanC enterococci with raw milk. Due to the low number of VRE isolates from mastitic milk and the inability to perform statistically robust analyses on VRE detection in individual milk samples, we decided to focus our statistical analysis on comparing the antimicrobial resistance profiles of mastitogenic E. faecalis and E. faecium, which were the most commonly identified species in the milk samples.

New response: Thank you for your comment. The prevalence of VRE in bulk milk that was obtained suggests that raw milk could be a potential source of vancomycin-resistant enterococcal exposure for humans. However, conducting a robust risk analysis of the consumption of raw milk and identifying possible associations by logistic regression between different parameters and the probability of milk carrying vancomycin-resistant enterococci was beyond the scope of this paper. We recognize that this could be a limitation of our paper. However, the low number of VRE isolates from mastitic milk, as well as the lack of information regarding why the analyzed milk was sampled (non-random sample selection), prevented us from conducting further analyses relating to risk assessment. On the other hand, we believe that readers may be interested to know whether there was an association between MDRS strains and the species of enterococci isolated, which we have reported in the paper.

Reviewer 3 Report

Comments and Suggestions for Authors

Comment 2: Line 101: Please provide reference or surveillance/epidemiological data on infections/outbreaks with Enterococci linked to consumption of raw (non-pasteurized) milk. Also, documentation of cases of Enterococci illness among farm personnel milking cows, veterinarians, and/or farmers.

Response 2: Thank you for pointing this out. Unfortunately, to the best of our knowledge, there aren’t data on infections/outbreaks with Enterococci linked to consumption of raw (non-pasteurized) milk or documentation of cases of Enterococci illness among farm personnel milking cows, veterinarians, and/or farmers.

 New Comment: Please include sentence in the paper indicating “…no data on infections/outbreaks with Enterococci linked to consumption of raw (non-pasteurized) milk or documentation of cases of Enterococci illness among farm personnel milking cows, veterinarians, and/or farmers”

Comment 4: Line 195: What was the incidence of mastitis caused by Enterococci on farms in which bulk tank samples were positive. None to low incidence may suggest the isolates were environmental origin and not associated with sub-clinical or clinical mastitis. I know that this is somewhat addressed on line 205-208. Wondering what percentage of cases of mastitis on those farms were linked to Enterococci infections.

Response 4: Thank you for pointing this out. Unfortunately, we do not have specific information on these data, as bulk milk samples submitted for quality analysis are identified only by barcodes. This makes it very difficult to determine the percentage of mastitis cases present in the farms submitting these samples.

New Comment: So, the barcode contains no information about the farm the sample was collected from.  How do the results get reported back to the farm? It seems you could contact the farm and they must keep records on incidence of mastitis particularly in reference to treatment of cases of clinical mastitis.  The farm would need to know what cows were treated with antibiotics to ensure the milk did not go into the bulk tank.

 Comment 5: Line 246: Were cattle treated with any of the antibiotics prophylactic or therapeutic?

Response 5: Thank you for your comment. Unfortunately, we don’t have information about these data.

New Comment: All farms maintain records of animals treated with antibiotics, particularly for cases of clinical mastitis.  This information is so important to the study considering the study evaluated susceptibility of isolates to antibiotics. Perhaps an email or phone call could be made to the farm operation and the information obtained.  If the authors cannot obtain the information then some type of statement should be included indicating antibiotic use by farm operations was not determined.  

Comment 7: Line 284: Provide greater information on sample collection.  For example, for individual samples were the cow’s udder washed and teats dried prior to collection. Were teat ends swabbed with alcohol swab prior to expression of milk into the collection tube. Were several streams of milk expressed prior to collection of the sample. Were samples taken prior to milking or after removal of the milking machine. 

Response 7: Thank you for pointing this out.  Unfortunately, we don’t have information about sample collection, because we just received the samples to test, and we are not involved in the sampling. Our lab consistently recommends following proper hygiene practices, but we were unable to verify whether these recommendations were actually implemented.

Comment: So, there is no way for you to contact the farms and ask what the procedure was for collecting milk.  Was there any visual oversight or training of the workers collecting the samples? Without this information it makes the research less impactful.  The bacteria in the individual milk samples could come from workers, fecal material, soil, etc and have no relation to the mammary gland. This may suggest that the results of the individual milk samples are meaningless. Collectively, the research presented may simply suggest that better on-the-farm hygiene practices are required to prevent Enterococci from contaminating bulk tank milk.

Author Response

Responses to Reviewer 3.

Comment 2: Line 101: Please provide reference or surveillance/epidemiological data on infections/outbreaks with Enterococci linked to consumption of raw (non-pasteurized) milk. Also, documentation of cases of Enterococci illness among farm personnel milking cows, veterinarians, and/or farmers.

Response 2: Thank you for pointing this out. Unfortunately, to the best of our knowledge, there aren’t data on infections/outbreaks with Enterococci linked to consumption of raw (non-pasteurized) milk or documentation of cases of Enterococci illness among farm personnel milking cows, veterinarians, and/or farmers.

New Comment: Please include sentence in the paper indicating “…no data on infections/outbreaks with Enterococci linked to consumption of raw (non-pasteurized) milk or documentation of cases of Enterococci illness among farm personnel milking cows, veterinarians, and/or farmers”

New Resonse: Thank you for your comment. The text was revised according to the Reviewer’s suggestion (lines 105-107).

Comment 4: Line 195: What was the incidence of mastitis caused by Enterococci on farms in which bulk tank samples were positive. None to low incidence may suggest the isolates were environmental origin and not associated with sub-clinical or clinical mastitis. I know that this is somewhat addressed on line 205-208. Wondering what percentage of cases of mastitis on those farms were linked to Enterococci infections.

Response 4: Thank you for pointing this out. Unfortunately, we do not have specific information on these data, as bulk milk samples submitted for quality analysis are identified only by barcodes. This makes it very difficult to determine the percentage of mastitis cases present in the farms submitting these samples.

New Comment: So, the barcode contains no information about the farm the sample was collected from.  How do the results get reported back to the farm? It seems you could contact the farm and they must keep records on incidence of mastitis particularly in reference to treatment of cases of clinical mastitis.  The farm would need to know what cows were treated with antibiotics to ensure the milk did not go into the bulk tank.

New response: we would like to clarify that bulk milk samples and mastitic milk samples follow two separate processing paths. Bulk milk samples are identified only by a barcode, and the corresponding identification data are stored in dedicated programs that we do not routinely access. In contrast, mastitic milk samples are assigned identification numbers within our laboratory. Both types of samples—those from farms submitting bulk milk and those related to mastitis—receive the results of the requested analyses (e.g., somatic cell count and bacterial count for bulk milk; identification of mastitis agents and corresponding antibiograms for mastitic milk) by an automatic transmission to the farmers. However, the results are stored on two separate servers. Due to this separation, as well as the high volume of samples processed, it is extremely difficult to determine whether farms with vanC-positive bulk milk samples also have cases of mastitis associated with vanC enterococci. This also why there is a huge variability, in terms of frequency and number of samples, among the farms in sending samples for diagnostic purposes. Therefore, we are unable to provide this information. 

Comment 5: Line 246: Were cattle treated with any of the antibiotics prophylactic or therapeutic?

Response 5: Thank you for your comment. Unfortunately, we don’t have information about these data.

New Comment: All farms maintain records of animals treated with antibiotics, particularly for cases of clinical mastitis.  This information is so important to the study considering the study evaluated susceptibility of isolates to antibiotics. Perhaps an email or phone call could be made to the farm operation and the information obtained.  If the authors cannot obtain the information then some type of statement should be included indicating antibiotic use by farm operations was not determined.  

New Response: Thank you for pointing this out. Added, lines 293-296

Comment 7: Line 284: Provide greater information on sample collection.  For example, for individual samples were the cow’s udder washed and teats dried prior to collection. Were teat ends swabbed with alcohol swab prior to expression of milk into the collection tube. Were several streams of milk expressed prior to collection of the sample. Were samples taken prior to milking or after removal of the milking machine. 

Response 7: Thank you for pointing this out.  Unfortunately, we don’t have information about sample collection, because we just received the samples to test, and we are not involved in the sampling. Our lab consistently recommends following proper hygiene practices, but we were unable to verify whether these recommendations were actually implemented.

New Comment: So, there is no way for you to contact the farms and ask what the procedure was for collecting milk.  Was there any visual oversight or training of the workers collecting the samples? Without this information it makes the research less impactful.  The bacteria in the individual milk samples could come from workers, fecal material, soil, etc and have no relation to the mammary gland. This may suggest that the results of the individual milk samples are meaningless. Collectively, the research presented may simply suggest that better on-the-farm hygiene practices are required to prevent Enterococci from contaminating bulk tank milk.

New response: Unfortunately, we cannot provide this kind of information, and we specified it in lines 293–296. Our lab provides to all customers (farmers, vets) specific guidelines for correct sampling and this has improved over time the quality of samples we receive for microbiological examination. However, we have not specific information about the procedures used by the single farms. About this point, the authors believe that this limitation does not negatively impact the results. In cases of poor sampling practices, contamination would typically be evident through the presence of mixed flora on the culture plates. Conversely, mastitis cases often result in pure colture. Moreover, enterococci—even when involved in mastitis—are environmental bacteria, likely originating from the environment, humans, or other external sources.

Round 3

Reviewer 1 Report

Comments and Suggestions for Authors

most of the responses goes to limitation.